# The Role of the Versius Surgical Robotic System in the Paediatric Population

**DOI:** 10.3390/children9060805

**Published:** 2022-05-30

**Authors:** Ewan M. Brownlee, Mark Slack

**Affiliations:** 1Department of Paediatric Surgery and Urology, Southampton Children’s Hospital, University Hospital Southampton, Southampton SO16 6YD, UK; ewan.brownlee@uhs.nhs.uk; 2Clinical School, University of Cambridge, Cambridge CB2 2QQ, UK; 3CMR Surgical Ltd., 1EBP, Milton Rd, Cambridge CB24 9NG, UK

**Keywords:** robot-assisted surgery, laparoscopic surgery, paediatric surgery, Versius surgical robotic system

## Abstract

The uptake of robot-assisted surgery has continuously grown since its advent in the 1990s. While robot-assisted surgery is well-established in adult surgery, the rate of uptake in paediatric surgical centres has been slower. The advantages of a robot-assisted system, such as improved visibility, dexterity, and ergonomics, could make it a superior choice over the traditional laparoscopic approach. However, its implementation in the paediatric surgery arena has been limited primarily due to the unavailability of appropriately sized instruments as per paediatric body habitus, therefore, requiring more technologically advanced systems. The Versius surgical robotic system is a new modular platform that offers several benefits such as articulated instruments which pass through conventional 5 mm ports, compact arms for easier manoeuvrability and patient access, the ability to mimic conventional port placements, and adaptive machine learning concepts. Prior to its introduction to paediatric surgery, it needs to go through a careful pre-clinical and clinical research program.

## 1. Introduction

Robot-assisted surgery is the paradigm of the most developed version of minimal access surgery (MAS) that has been introduced to expand the competencies of surgeons while addressing the challenges and complications associated with endoscopic surgery [1,2]. Surgeons performing traditional MAS are more prone to experience muscle injury and fatigue, especially in the upper limbs, neck, and head when compared with surgeons performing open surgery [3]. Robot-assisted surgery intends to address these issues by improving ergonomics and, therefore, lowering the physical stress for surgeons while still providing patients with the benefits of MAS [4,5]. Since the advent of the first robotic system in 1994, robot-assisted surgery has been continuously evolving with increasing precision and effectiveness and has now been adopted by a variety of surgeons all over the world [6,7].

Robot-assisted surgery has also been utilized in the paediatric population for the past two decades, with numerous reported clinical benefits including less postoperative pain, shorter hospital length of stay, reduced operative trauma, minimal scarring, less postoperative pain, reduced opioid use, reduced risk of infection, and improved cosmetic outcomes [6,8]. Following the first described use of robot-assisted surgery in children in 2001 [9], its applications in the paediatric population have expanded to include multiple procedures across specialties such as urology, cardiothoracic surgery, and general paediatric surgery for patients ranging in age from infants to adolescents [10], with studies indicating its safety and efficacy in children [11,12]. 

Despite the fact that the surgical robotic marketplace is expanding, the da Vinci Surgical System (Intuitive Surgical, Sunnyvale, CA, USA) and the Senhance System (Asensus Surgical, Durham, NC, USA) are currently the only robotics systems authorized for paediatric usage [10]. In 2000, the da Vinci Surgical System was approved by the United States Food and Drug Administration (FDA, Silver Spring, MD, USA) for urology, gynaecology, cardiothoracic, colorectal, head and neck, and general surgery, both for the adult and paediatric population [13]. However, technological limitations of the Da Vinci Surgical Systems, including size of the surgical robot, cost of instrument, and lack of haptic feedback, has led to the development of new robotic systems that are primarily working to address these issues [13,14,15]. The Versius robotic system is a new, smaller, modular open console system with instruments able to pass through 5 mm laparoscopic ports [4,16]. With compact robotic arms and easier manoeuvrability, as well as adaptive machine learning concepts, this may bring robot-assisted surgery to a next level in the paediatric population [13,17]. We wondered whether this new robotic system may have distinct advantages in the paediatric population, potentially increasing the uptake of robotic paediatric surgery.

## 2. Advantages of Robotic Surgery in Paediatrics 

The current armamentarium for paediatric surgery includes open surgery (traditional) and MAS (conventional laparo-thoracoscopic surgery and robot-assisted surgery). The most advanced surgical tool (i.e., robotic-assisted surgery) improves laparoscopic surgical skills by its more refined hand–eye coordination, fine motor movement, wristed instruments, superior suturing skills, improved vision, better dexterity, more precise movements, as well as greater precision in dissection [6]. 

The robotic system consists of consoles that allow the surgeons to sit while operating, joystick controls that compensate for the fulcrum effect caused by the use of laparoscopic equipment, as well as 3D glasses that offer magnified views of the surgery [18]. Robotic systems are especially designed so as to mimic human wrist movements, allowing seven degrees of freedom of movement, which can be advantageous for complex surgeries, especially those in the relatively confined working spaces of paediatric patients and difficult anatomical areas [19,20]. 

Design features of robotic devices include motion scaling, stereoscopic vision, greater optical magnification, instrument indexing, increased instrument tip dexterity, operator-controlled camera movement, tremor filtration, and elimination of the fulcrum effect [21,22,23]. These technological advancements offer improvements to traditional MAS that help in exceeding human threshold capacities for surgery within the limited operating workspaces in children [24]. 

Improved dexterity could be particularly useful in paediatric surgery, where the smaller body habitus of neonates, infants, and young children makes some anatomic areas more difficult to access. Moreover, robotic systems have motion scaling, thereby, reducing the scale of the surgeons’ movements by 5:1, and hence allowing precise movements in narrow and small spaces [10]. 

The highly magnified 3D images provided by the robotic surgeon console permits a level of visibility not possible with open and traditional laparoscopic procedures [10]. Robotic devices can magnify images up to 15 times, allowing for better depth perception and surgeon control [20,25]. Robotic cameras also enable tremor filtration plus operator-controlled views, allowing for more precise and steadier vision which is critical in paediatric surgery as anatomy visualization in paediatric patients is challenging because of their small size [6,10]. Although there are no significant differences in the success rates of open and robotic surgery, the latter has ergonomic benefits for surgeons as well as potentially reducing intra- and post-operative complications [15,26]. Overall, robot-assisted surgery has numerous benefits for the paediatric patients, such as reduced operative trauma, decreased scarring, less postoperative pain, reduced opioid use, fewer post-operative complications, less bleeding and need for transfusions, less risk of infection, shorter hospital length of stay, and speedy return to daily activities, which also benefits parents [20,23,27]. When evaluating the cost of robotic surgery in the paediatric population, in addition to other benefits, reduced parental time off work, patient time off school, and patients’ emotional benefits should also be considered [6].

## 3. Special Considerations for Robotic Surgery in the Paediatric Population

The technical capabilities of robot-assisted surgery are particularly advantageous in challenging paediatric surgical cases that requires precise dissection and reconstruction, especially procedures requiring a lot of suturing. However, there are still a number of challenges faced while performing robot-assisted surgery in paediatric patients. 

The size of the robotic system is the principal concern for paediatric patients. This is critical as it may hamper rapid intraoperative access to the patient, putting the anaesthetic team at a disadvantage [28,29]. 

Port size is also a significant factor for paediatric surgery. The majority of paediatric laparoscopic surgery is done with 5 mm or even 3 mm ports, using larger port sizes very infrequently compared with adult practice. The relatively small size of the patient makes the port size more important, in particular when some open operations can be done through very small incisions. This has been one of the major factors slowing the uptake of robotic-assisted surgery in children [30]. Port placement is another major challenge in paediatric patients for two main reasons. Compared with adults, the abdominal wall of children is more lax; and hence, there is a greater chance of injury to major vessels or viscera during the insertion of the ports [1,9]. In addition, due to less intracorporal space in children because of their smaller abdomen, port placement must be pre-determined carefully in order to prevent collision of arms when operating. This is technically challenging in younger children, particularly in new-borns [1,9]. 

Next, the insufflation pressures which are age dependent should also be considered in paediatric surgery. Generally in infants <2 years, intra-abdominal pressures between 8 and 10 mmHg are recommended, whereas an optimum range of 10–12 mmHg are suitable for children between 2 and 10 years [1,9]. In addition, there is compromised visibility in children because of lesser gastric emptying time, thereby, causing small bowel distention [1,9]. Both of these can limit operative space, further contributing to the challenge of performing MAS in the paediatric population. 

Robot-assisted surgery is still not the standard, particularly in the paediatric population, and thus technological advancements are continuing to improve the currently available robotic systems [20]. Table 1 summarizes the advantages and limitations of robot-assisted surgery.

## 4. Applicability of Robotic Surgery in the Paediatric Population

Robotic surgery has been utilised in several paediatric surgical subspecialties, such as urology, oncology, general surgery (gastrointestinal hepatopancreatobiliary), thoracic, and otorhinolaryngology [10,20,31]. Robot-assisted technology offers support to the paediatric surgeon by improving dexterity and precision of movements, hand–eye coordination, ergonomics, and visualization [21]. 

### 4.1. Paediatric General Surgery 

Since the first reported robotic-assisted paediatric general surgery in 2001 [9], a range of surgeries have been performed and many studies have been conducted to establish the feasibility of robotic surgery in paediatric general surgery. Lehnert et al. [32] published a prospective research paper in 2006 comparing operation time in children undergoing traditional laparoscopic vs robot-assisted Thal fundoplication. The study concluded that the robot-assisted group took more operative time, but the dissection of the hiatal region was completed faster (34%). The study also showed that, in terms of procedural outcomes, the robotic technology outperformed traditional laparoscopic approaches. However, the benefit of reduced operative time was offset by the longer set-up time. Another study in neurologically impaired children undergoing robot-assisted laparoscopic Nissen fundoplication found particular benefits in children with adhesions from previous abdominal surgery, previous gastrostomy, or failed primary fundoplication [33]. Robotic fundoplication has also successfully been reported by others [34]. A case report, published in 2007 on the robotic repair of a Bochdalek congenital diaphragmatic hernia, concluded that the repair can be successfully performed even in small neonates [35]. Laparoscopic Morgagni hernia repair is another procedure successfully performed via robotics [36,37]. The authors stated the successful completion of two cases, one in a 5 year old and the second in a 23 month old, and another in a 5 year old with surgeons reporting increased dexterity provided by the robot-assisted system.

### 4.2. Paediatric Cardiothoracic Surgery 

Various case reports and studies have also described paediatric cases utilizing the robot in cardiothoracic surgery. In 2007, a case report published by Robinson et al. described the robotic division of an uncommon variant of a right aortic arch in a 6-year-old symptomatic boy. On follow up, the child had recovered exceptionally well [38]. Another case report by Baird et al. described the first case of an entirely endoscopic/robotic closure of an atrial septal defect (ASD) in a child, utilizing the da Vinci system and hypothermic fibrillation. The research group concluded that the benefits of endoscopic/robotic ASD repair combine enhanced vision and improved precision and visualization with a reduced amount of pain along with better cosmetic results [39]. Meehan et al. [40] described the first report of experiences using a robotic pulmonary resection procedure even in small children and confirmed that these procedures can be performed effectively even in small infants. In the same year, Meehan et al. [41] stated their experiences in children with robotic resection of mediastinal masses, concluding that robotic surgery is safe and effective in children for resecting solid mediastinal chest masses.

### 4.3. Paediatric Urology Surgery 

Amongst paediatric disciplines, urology is the subspecialty in which robot-assisted surgery is most utilised. General advantages have been shown to be reduced post-operative pain and analgesia requirements, shorter hospital stays, and improved cosmesis compared with open surgery [3,42,43]. The most commonly performed robot-assisted procedure in children is the pyeloplasty. This has been shown to have better outcomes and shorter procedural times compared with both laparoscopic and open approaches [44,45]. Analgesia requirements and hospital stays have both been shown to be reduced compared with the open approach [46]. Other common upper tract procedures (nephrectomy and heminephrectomy) have also been shown to be successful when performed using robotic assistance [47,48]. Ureteric surgery (ureteric reimplantation and ureteroureterostomy) has been shown to have similar very high success rates to the open procedures, but with reduced post-operative pain and shortened hospital stays [49,50,51,52]. 

Probably the area which has the most potential for significant development by the use of robotic technology is bladder reconstructive surgery (Mitrofanoff (appendicovesicostomy) and bladder augmentation (ileocystoplasty and ureterocystoplasty)). The majority of these procedures are currently performed open, often with long post-operative stays and potential issues with ileus and adhesions. These procedures are technically very challenging using traditional laparoscopic surgery, but the enhanced movement of robotic-assisted surgery should allow more of these procedures to be performed using MAS. There is evidence that, when performed using robotic assistance, these procedures have comparable outcomes to open surgery, but with significantly reduced length of hospital stay [53,54]. 

### 4.4. Paediatric Emergency Surgery

There seems to be very little published use of robotic surgery in paediatric emergencies, which may relate to the simple nature of common emergency operations, such as appendicectomy, where robotic assistance may not be felt to be necessary, or the perceived tactile need in emergency neonatal laparotomies for NEC or volvulus. We would hypothesise that this may relate to the limited uptake of paediatric robotics in general, meaning there is not only a relatively small pool of surgeons using robotic surgery, but also, therefore, a smaller cohort of theatre staff familiar with robotics, especially out of regular working hours. Robotic assistance may prove particularly useful in urgent procedures such as the repair of traumatic or congenital diaphragmatic hernia or duodenoduodenostomy, for example.

## 5. Surgical Robotic Technologies

Robotic surgical platforms are becoming more popular, and newer technologies are being developed continually [18]. Currently, the only robotic systems approved for use in the paediatric population are the da Vinci Surgical System, the first robotic surgery system to be approved by the United States FDA for intra-abdominal surgery, as well as the Senhance by Asensus [15]. Despite technological advancements, the da Vinci robotic system still has several limitations [10,15]. Its use in paediatrics has some limitations. The size of the system and the amount of working space needed by the device internally, for the usage and articulation of the instruments, is one of its restrictions [15]. This surgical system is also associated with longer operation and anaesthesia times, which is a matter of concern when considering a paediatric population. Moreover, the robot set-up and docking time is higher in the earlier systems, which causes the patients to remain entirely paralyzed until the robot is docked [14]. These limitations pose additional challenges for paediatric patients.

### Versius—A New Generation Robotic System

The Versius surgical system (CMR Surgical, Cambridge, UK; Figure 1) is a new robotic platform that launched its first United States training program recently in collaboration with Florida Hospital Nicholson Centre [15]. The Versius system is a small, modular open console surgical robot designed for laparoscopic, gynaecological, upper GI, colorectal, thoracic, and urological surgeries [18]. This surgical system has successfully shown its ability to perform needle-driving suturing, electro-surgery, and tissue manipulation during the Cadaveric trials performed at the Evelyn Cambridge Surgical Training Centre [4]. Likewise, the applicability of the Versius system for transanal as well as mesorectal tumour excision has been assessed [55]. The Versius surgical system has also been shown in various preclinical studies to successfully complete a variety of urological, renal, gynaecological, as well as general surgical procedures [17,56,57]. The Versius system has been used extensively across a range of adult surgeries including gynaecology, urology, colorectal surgery, general surgery, and thoracic surgery. 

The Versius system has also been tested to perform robotic reconstructive procedures, including intracorporal suturing and knot tying, in small boxes simulating paediatric-sized cavities [16]. These were achievable in boxes with a volume as small as 106 mL (less than 5 cm in each dimension).

With several unique and advanced features, the Versius system provides several benefits. The Versius robotic system offers wristed instruments used through 5 mm ports with shorter articulating jaws, thereby making robotic interventions in new-borns appear achievable [16]. It is relatively small, more flexible, and versatile, allowing it to accomplish a wider range of operations. The Versius system has modular robotic arms, thus adding arm-positioning flexibility in configuration and the choice of using two or three operating arms as required [55]. This modular design of the system further allows flexibility of positioning in the operative theatre, while providing several robotic arms connecting instruments including scissors, graspers, needle drivers, and electrocautery instruments [18]. The system’s mobility and modest size makes it easy to adopt to hybrid manual–robotic procedures, thereby reducing the conversion time from robotic to open surgery in case of an emergency [4]. The system’s instrument and visualization arms are each attached to their own wheeled cart, forming a compact and movable bedside unit (BSU), thereby providing maximum flexibility in operation theatres [57]. Versius offers an open-console design, thus allowing surgeons to stand or sit, and have easy communication with the team [13]. This also facilitates teaching and training. The surgeon controls the robotic arms of the system with joystick-like controllers at the console, wearing 3D (HD) glasses, and viewing the monitor [57]. A comparison between the currently approved da Vinci and newly developed Versius is given in Table 2. 

The Versius design, throughout its development, has employed end-user feedback to refine the design so as to ensure that it meets user needs [58]. The Versius system allows a variety of port placements providing adequate surgical access and reach; this flexibility enables surgeons’ to effectively transfer their preferred laparoscopic port placements, when desired, for use with the robotic system. This again helps in reducing the learning curve associated with robot surgery. It may also introduce the option of placing port incisions in more cosmetic positions. Moreover, the use of standard, disposable 5 mm trocars for the operating arms increases the versatility of the system [17]. The Versius system design allows the surgeon to adopt a more neutral position. This together with the articulated instruments provides improved ergonomics and hopefully a reduction in MAS-associated injuries to the surgeon [4]. 

## 6. The Future of Robotic Surgery in the Paediatric Population

With evolving technology and the need for more compact robotic platforms, the future of robotic surgery will undoubtedly result in improved instrumentation. Reconstructive surgery (such as esophageal, intestinal, and renal tract anastomoses), which necessitate a delicate and precise approach, will benefit immensely from these advances. The paediatric and neonatal patient must be at the forefront of research into the future of robotic surgery. 

## 7. Limitations

Whilst the Versius robotic system would appear to have significant potential benefits in the paediatric population, at the time of writing it has not yet been used in children. In order to confirm the benefits of this new system in paediatric practice, careful pre-clinical and clinical research will be required to ensure this can be introduced safely and effectively.

## 8. Conclusions

The applicability of robot-assisted surgery is gradually making its way among the paediatric surgical population following the success and proven benefits in the adult population. Robotic surgeries conducted in paediatric populations have demonstrated its safety, feasibility, and efficacy across a wide range of procedures in children. The Versius robotic system, with its small size and other unique attributes, may be able to perform a range of paediatric surgery. However, the system must first undergo extensive pre-clinical and clinical studies before entering widespread use.

## Figures and Tables

**Figure 1 children-09-00805-f001:**
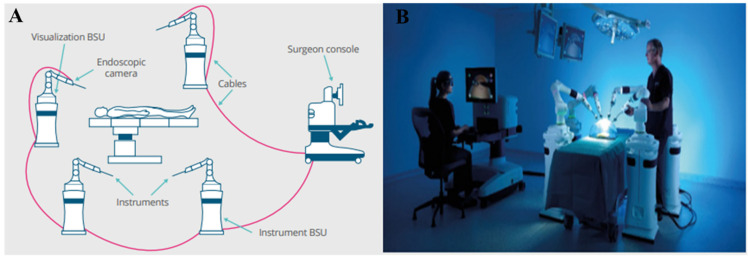
Overview of the Versius surgical robotic system: (**A**) schematic overview of the Versius system and (**B**) the operational set-up image of the Versius system. BSU, bedside unit.

**Table 1 children-09-00805-t001:** Advantages and limitations of robot-assisted surgery.

Advantages	Limitations
3D, high-resolution visualization	Limited number of the staff who are familiar with the system
Instruments provide greater range of motion	Size of robot which can limit access to the patient
Motion scaling and tremor control	Robot set-up and docking times
Intuitive movement	High capital and maintenance costs
Seven degrees of freedom	Limited instrumentation selection for paediatric patients
Decreased postoperative opioid needs as well as lower pain scores	No standardized way to place trocars
Shorter hospital length of stay	Physiologic effects related to insufflation pressure
Improved ergonomic features	High cost of disposables
Improved cosmetic outcomes	Physiologic effects related to absorption of carbon dioxide

**Table 2 children-09-00805-t002:** Differences between da Vinci and Versius robotic surgical platforms.

	Da Vinci Xi	Versius
Manufacturer	Intuitive surgical	CMR surgical
Country of origin	United States	United Kingdom
FDA status	Approved in USA	Pending in USA
Console/workstation	Remote (closed)	Remote (open)
Arm configuration	Single cart	Modular
3D viewing	Viewfinder	Passive 3D glasses
Number of arms	4 max	4
Effector arm diameter	8 mm	5 mm
Effector arm lifespan	10	13 (projected)
Ability to employ trocars	No	Yes
Foot-pedal control	Yes	No
Ability to operate in two fields	No	Yes (experimental)

## Data Availability

Not applicable.

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
