# Peer review of "The Role of the Versius Surgical Robotic System in the Paediatric Population"

_children, 2022, doi:10.3390/children9060805_

Round 1

Reviewer 1 Report

At the outset, I would like to congratulate the team headed by Mark Slack et al., for their efforts in creating a new Robotic system and revolutionizing the field of Robotic surgery. Being a Pediatric Robotic Surgeon, I really enjoyed reading this article. In the current literature, numerous single-center experiences have been published on robotic surgery in children.

I have several comments that will improve the overall scientific quality of this narrative review.

Introduction: The first study on the comparison of ergonomic risk in robot-assisted laparoscopic versus conventional laparoscopic surgeries in children was published by a center in India (10.1089/lap.2021.0471). I would advise the authors to cite the work.

Since then, we have been comparing the ergonomic risk in different surgical procedures. Being a proponent of pediatric Robotic surgery, I must say that it is here to stay.

Please write your hypothesis in 1-2 lines at the end of this section. What prompted you to conduct this narrative review?

Sections no 2 and 3 include the advantages of robotic surgery (in general). I would advise the authors to include them in a single section as the “Advantages of Robotic Surgery in Children”

 Special considerations of robotic surgery in children: There are many issues that can be highlighted:

A recently developed concept is the utilization of stealth port placement. This concept of insertion of ports at non-standard sites is particularly useful in children, where the issues of space constraints arise quite frequently. We have been using this concept in a number of pediatric robotic surgical procedures. I strongly believe, the Versius system can outscore other robotic systems as it allows the use of 5 mm ports instead of standard 8 mm ports. For example, we have been operating on children with choledochal cysts utilizing 5 mm ports- all at the suprapubic position. This not only conceals the port scars but allows a pan-view of the adjoining structures.

Another important consideration in children is the “burping”. This maneuver allows the creation of more space inside the abdominal cavity, which is particularly important in children. 

Applicability of robotic surgery in the pediatric population:

Again, I would advise the authors to cite the recently published comprehensive experience on pediatric robotic surgery (10.1089/lap.2021.0183). The paper includes a variety of procedures performed using robot assistance.

I would suggest the authors to avoid duplication of text- please don’t use the phrases- modular arms, small-sized ports, etc., multiple times where it is not required.

The obvious reasons why the Versius system proves to be advantageous as compared to the other robotic systems in pediatric robotic surgery is because of smaller-sized ports (5 mm) and modular arms. The surgeon can restrict himself/herself to two or three arms (as per the need). In addition, the provision of small-sized ports also allows easy introduction of stealth port placement.

Limitations: Please write a paragraph on the limitations before the conclusions section.

Author Response

Thank you very much for your comments.

We have added citations for both of your papers as requested.

We have added a further sentence to the hypothesis at the end of the introduction to make it more explicit as requested.

We have combined sections 2 and 3

We have added a comment on the flexibility of positioning of the Versius arms to potentially facilitate more cosmetic siting of wounds

‘Burping’ does not seem to be needed with the Versius system so we have not commented on this in the paper

As requested, we have reduced duplication of the phrases ‘smaller sized ports’ and ‘modular arms’ as much as possible, but as you have mentioned, these are the principle potential benefits of the Versius system.

We have added a Limitations Section (section 7) before the conclusion

Reviewer 2 Report

The paper is a comprehensive review of the literature regarding the role of robotic surgery in the pediatric population, in particular, the Versius Surgical Robotic System. The article is well structured and it is written in a clear and concise manner. The advantages and limits of this method are presented in concordance with the specific features encountered in children.

The authors could add a paragraph regarding the possible use of robotic surgery in emergencies in pediatric surgery.

Author Response

Thank you very much for your comments.

We have added a section on the use of robotic surgery in emergencies in paediatric surgery (into section 4) as suggested. Unfortunately there do not appear to be published series of specific emergency robotic paediatric surgery procedures to be able to reference – we would hypothesise that this may relate to the relatively limited uptake of paediatric robotics in general, and therefore there is not only a small pool of surgeons using robotic surgery, but also a smaller cohort of theatre staff familiar with robotics.

Reviewer 3 Report

I would like to congratulate the authors for their work describing the new Versius robotic sistem and its use in the paediatric population.

However, I would like to add a few comments:

  1. Can the authors provide more information regarding the tehnical specifications of the instruments used for paediatric surgery? Are they different from adult population/ what size/ costs/ etc...
  2. When comparing the Versius system with the daVinci Intuitive robot, please provide the costs in comparison: cost of aquisition of system, maintainance, disposable and specific instruments, etc... It would be interesting to see the cost balance, especially for practitioners who intend to aquire a robotic system and now have the choice between daVinci and Versius.

Thank you!

Author Response

Thank you very much for your comments.

Regarding point 1, the specification of the instruments would currently be the same for paediatric surgery as for adult patients. We hypothesise that the smaller 5mm instruments would be suitable for children without needing specific changes. We plan further clinical research to further explore this. Additional instruments will be added in due course such as advanced energy systems and enhanced visualisation.

Regarding point 2, we did not cover cost comparisons within this paper, partly due to the large number of variables and different pricing schemes, partly to maintain clinical focus on the potential paediatric application of the Versius robot.  The Versius system is purchasable through either a capital purchase or leasing model. CMR’s aim would be that with increased utilisation the cost would be comparable to conventional laparoscopic surgery.

Round 2

Reviewer 1 Report

I would like to congratulate the authors on their work. In the revised manuscript, all my comments have been addressed. The overall scientific quality of the manuscript has improved significantly.

Reviewer 3 Report

No further comments.